# Abrupt drainage basin reorganization following a Pleistocene river capture

Niannian Fan[1,2,3], Zhongxin Chu[4], Luguang Jiang[5], Marwan A. Hassan[2], Michael P. Lamb[6] & Xingnian Liu[1]

River capture is a dramatic natural process of internal competition through which mountainous landscapes evolve and respond to perturbations in tectonics and climate. River capture may occur when one river network grows at the expense of another, resulting in a victor that steals the neighboring headwaters. While river capture occurs regularly in numerical models, field observations are rare. Here we document a late Pleistocene river capture in the Yimeng Mountains, China that abruptly shifted 25 km$^2$ of drainage area from one catchment to another. River terraces and imbricated cobbles indicate that the main channel incised 27 m into granitic bedrock within 80 kyr, following the capture event, and upstream propagating knickpoints and waterfalls reversed the flow direction of a major river. Topographic analysis shows that the capture shifted the river basins far from topographic equilibrium, and active divide migration is propagating the effects of the capture throughout the landscape.

[1] State Key Laboratory of Hydraulics and Mountain River Engineering, College of Water Resource & Hydropower, Sichuan University, 610065 Chengdu, China. [2] Department of Geography, University of British Columbia, Vancouver, BC V6T1Z2, Canada. [3] State Key Laboratory of Loess and Quaternary Geology, Institute of Earth Environment, Chinese Academy of Sciences, 710061 Xi'an, China. [4] Key Laboratory of Submarine Geosciences and Prospecting Techniques of MOE, Ocean University of China, Laboratory for Marine Geology at Qingdao National Laboratory for Marine Science and Technology, 266100 Qingdao, China. [5] Institute of Geographic Sciences and Natural Resources Research, Chinese Academy of Sciences, 100101 Beijing, China. [6] Division of Geological and Planetary Sciences, California Institute of Technology, Pasadena, CA 91125, USA. Correspondence and requests for materials should be addressed to Z.C. (email: zhongxinchu@ouc.edu.cn)

River captures have attracted attention for more than a century[1–11], due to their significance in both landscape evolution[2,12,13] and ecology[14]. However, river captures are still poorly understood because they occur infrequently[1], and often lack clear field evidence[15–17]. Instead, studies of river capture often rely on numerical models[4–6,18–20], which reveal their importance in dramatically changing drainage network geometries, river flow directions, and the evolution of landscapes. However, with few exceptions[21], most documented river captures are ancient, ranging in age from 100 ka to more than 20 Ma[10–12,22–24], and their transient effects on landscape evolution, such as pulses of river incision and drainage basin reorganization, are difficult to determine from field observations[20,24].

Here we report on a Late Pleistocene river capture in the Yimeng Mountains, China (Fig. 1 and Supplementary Software 1 for Google Earth viewing), which is sufficiently recent that the

landscape is actively adjusting, and evidence of the capture is well preserved. Thus, this landscape provides critical insight into the mechanism of river capture and subsequent landscape adjustment.

## Results

**Study site**. The study area includes the upstream portion of the Chaiwen River and the Yihe River, which are main tributaries of the Yellow and Huaihe Rivers, respectively (Fig. 1). The Chaiwen River has a large bend, referred to as the elbow, where the river abruptly changes directions by ~300°, which separates the lower and upstream portions of the catchment. The Yihe River and the Chaiwen River, downstream of the elbow, follow the Shangwujing Fault (Fig. 1b), and the Tongyedian normal Fault bounds the study area to the southwest; these faults are not thought to be active during late Quaternary[25,26]. The study area also was unglaciated during the Quaternary[27], and sea level change had no direct effects on this area. Bedrock is generally uniform granite, and the climate is temperate semi-humid monsoon, with large storm induced summer and autumn floods.

The elbow of the upper Chaiwen River is unusual for a river network geometry, and in planview its flow direction is more in line with the Yihe River that flows to the northeast (Fig. 1b). There is a tributary to the Chaiwan River, the Daotang River, at the apex of the elbow and it is also directly in line with Yihe River in planview, but drains in the opposite direction (Figs. 1b and 2a). The longitudinal profiles (Fig. 1c) of the rivers indicate that the divide between the Daotang River and Yihe Rivers is low and flat (referred as a wind gap). This network geometry suggests that the upper Chaiwen River basin, upstream of the elbow, may have once drained to the Yihe River, and that the Chaiwen River subsequently captured this catchment area; if this hypothesis were correct, the paleo-Daotang River (dashed line, Fig. 1c) must have entrenched and reversed directions following the capture.

**Field measurements**. Several fluvial terraces abut the Daotang River, and we identified imbricated fluvial cobbles at three locations at elevations of 25, 13, and 3 m above the modern riverbed (D1, D2, and D3 in Fig. 2). River cobbles develop an imbricated pattern because of hydrodynamic stability during sediment transport, and their plane containing both the long and the intermediate axes tends to dip downward in the upstream direction[28]. The Daotang terrace cobbles preferentially dip downwards to southwest (Fig. 3 and Supplementary Data 1 for detailed information), indicting a paleo-flow direction to the northeast (Fig. 2a), which is opposite to the present-day river direction, supporting the river capture hypothesis. Silt deposit above the imbricated cobbles at D1 and sand deposit above the imbricated cobbles at D3 (Fig. 2a; Supplementary Fig. 2 and Fig. 3) have deposition ages of 82.53 ± 4.29 and 89.48 ± 3.03 kyr before present (Methods), respectively, suggesting that the river capture occurred less than ~80 kyr ago, and subsequently the Daotang River reversed flow directions and entrenched by up to 27 m into bedrock abandoning the terraces.

The longitudinal profile of the Chaiwen River is relatively steep close to the elbow and it contains one knickpoint (i.e., prominent series of waterfalls) with a height of 2.2 m located in the Chaiwen River 1.4 km upstream of the elbow (Fig. 2a, c). The knickpoint bounds the upstream end of an entrenched, bedrock inner gorge (Fig. 2 and Supplementary Fig. 1), suggesting that the upstream knickpoint propagation is responsible for inner gorge formation. We interpret that the knickpoint formed during the capture at the elbow, and has since propagated upstream in Chaiwen River at an average rate of at least 1.75 cm yr$^{-1}$ (i.e., 1.4 km in less than 80 kyr). Thus, the knickpoint indicates a propagating wave of

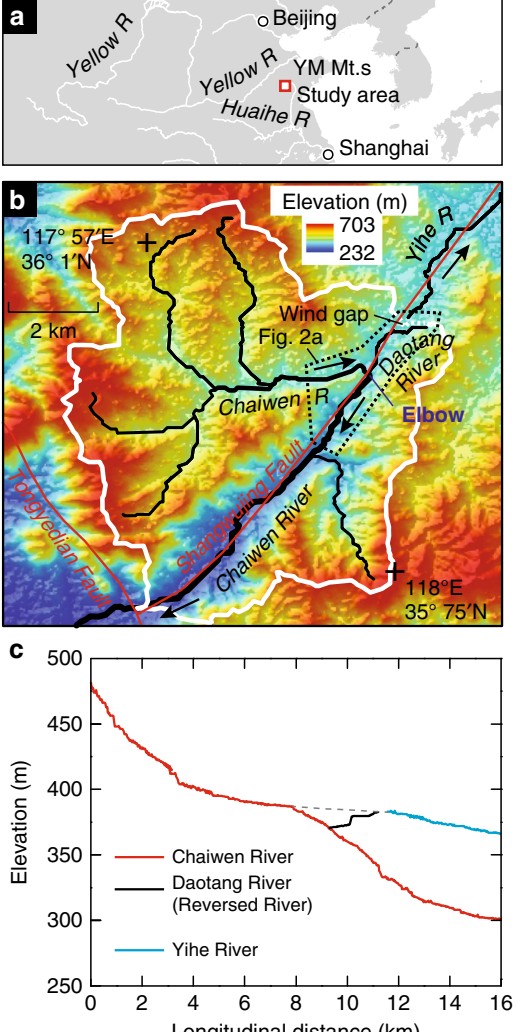

**Fig. 1** Overview of the study area. **a** Location of the study area in China. **b** Shaded relief map showing elevations within the study area; the white line is the boundary of the upstream portion of the Chaiwen River Basin, and the lowest saddle in the divide between Daotang and Yihe Rivers (wind gap) is indicated as a dashed line. The red solid lines are faults and the dashed area around the elbow is shown in detail in Fig. 2. **c** Longitudinal profiles of the Chaiwen, Yihe, and Daotang Rivers. The dashed line is a hypothesized longitudinal profile of the paleo-channel that connected the upper Chaiwen River to the Yihe River before the capture event, and entrenchment and reversal of the Daotang River. The data for **b** and **c** are from ref. [39]

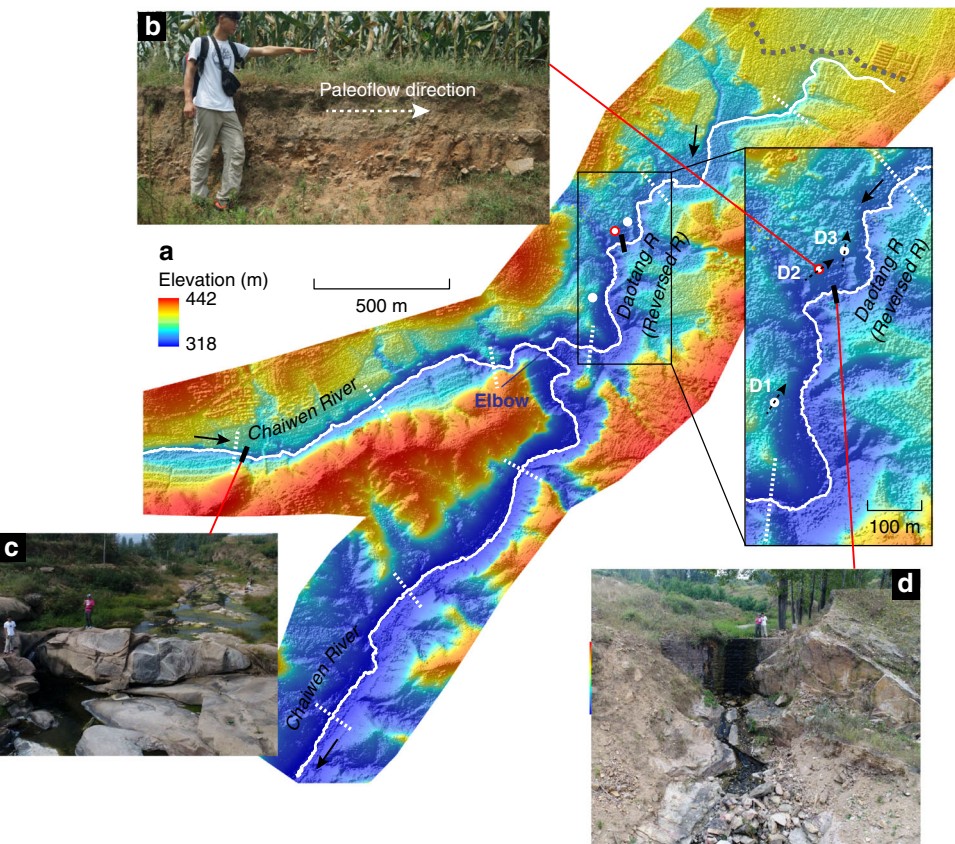

**Fig. 2** A close-up of the study area. **a** Shaded relief map with a resolution of 0.5 m covering the area close to the elbow produced using an Unmanned aerial vehicle (UAV) (see Methods). White dots represent locations for D1, D2, and D3 where the dip directions of imbricated cobbles were measured (see Fig. 3 for rose plots) and the median paleo-flow directions indicated by the cobbles are shown as black dashed arrows, opposite of the present flow direction of Daotang River. Black bars denote the location for the two waterfalls and gray dashed line denotes the wind gap. Dashed white lines indicate the traces of cross-sectional profiles shown in Supplementary Fig. 1. **b** A deposit in an old terrace looking west where the dip directions of the imbricated cobbles were measured (D2). The guy in the image provides a rough scale. **c** The 2.2 m high knickpoint in Chaiwen River. **d** The 10 m high knickpoint in the Daotang River

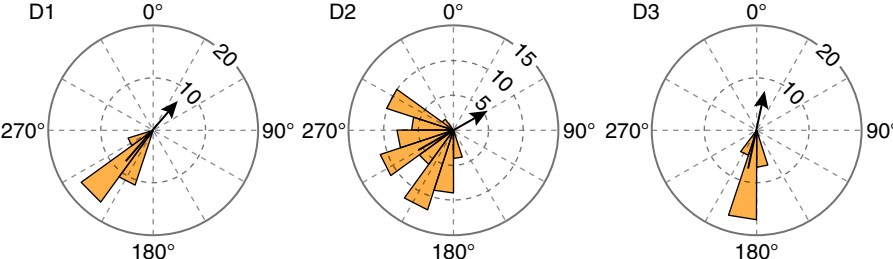

**Fig. 3** Dip directions of cobbles in three locations in terraces along the Daotang River indicate the paleo-flow direction was opposite to that of the present. Downward dip direction of the plane containing both the long and the intermediate axes of cobbles were measured, the azimuthal directions are reported in degrees. Dashed circles indicate the number of measurements (32, 68, and 34 cobbles were measured at each site, respectively). D1, D2, and D3 refer to the measurement locations shown in Fig. 2a. Black arrows show the inferred paleo-flow directions (opposite to the median dip direction of the cobbles)

incision due to the increased drainage area and the lowering of the base level of the Chaiwen River following the capture event[29–32].

Due to entrenchment of the Chaiwen River, we infer that a knickpoint also translated upstream the paleo-Yihe, reversing its flow direction and creating the Daotang River. This knickpoint is expressed as the wind gap between the Daotang River and the Yihe River and propagated upstream at rates greater than 2.38 cm yr$^{-1}$ (i.e., 1.9 km in less than 80 kyr). Although the wind gap must be the leading knickpoint formed initially at the elbow following the capture event, much of the entrenchment of the Daotang River, relative to the terraces, occurs within in an inner gorge downstream of a 10-m high waterfall, located 1.1 km downstream of the wind gap (Fig. 2a, d, Supplementary Fig. 1). The waterfall likely developed as the Daotang steepened in response to enhanced incision of the Chaiwen River following the capture. Because the wind gap itself has negligible drainage area, the downstream waterfall may reflect a threshold water discharge needed for significant bedrock erosion[33], or the larger discharge threshold necessary for waterfall erosion compared to fluvial incision[34,35].

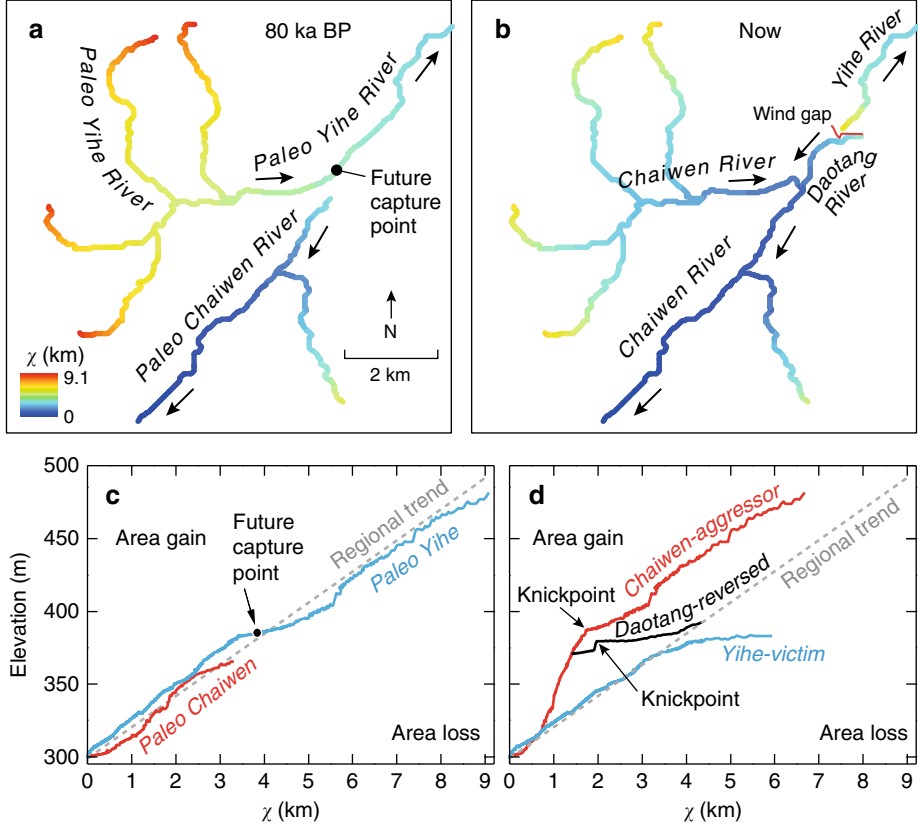

**Fig. 4** River network and longitudinal profile before and after river capture. **a** $\chi$ map for the river network before river capture. **b** $\chi$ map for the present river network. **c** $\chi$-elevation plot for the Paleo Chaiwen and Yihe Rivers before capture. **d** $\chi$-elevation plot for the present Chaiwen, Yihe, and Reversed Rivers. The data are from ref. [39]

**River profile analysis**. Being well preserved and relatively recent, the Chaiwen-Yihe capture offers a rare opportunity to reconstruct the topography and river network from before and after the event (Methods), which can be used to evaluate a recently proposed topographic metric, $\chi$, for drainage basin adjustment[2]. The $\chi$ parameter is a metric for weighted drainage area in a river network (Methods) and indicates, according to the stream-power model for river incision for areas with uniform distributed uplift rates and lithology, disequilibrium in river profiles due to transient adjustment[36]. Before the capture event, the $\chi$-elevation relations of both Paleo Chaiwen and Yihe Rivers generally fitted the regional trend, indicating no significant disequilibrium. However, headward extension of the Chaiwen River, following a linear path in $\chi$-elevation space, does intersect the eventual capture point (Fig. 4c), predicting an eventual capture event. This pathway is also along the trace of the Shangwujing Fault and more erodible rocks along the fault zone may have enabled the capture.

After the capture, the Chaiwen River gained 25 km² in drainage area at the expense of the Yihe River. Overall, this caused both river profiles to be far out of equilibrium in their headwaters (Fig. 4d), which is also evidenced by the migrating knickzones in the Chaiwan and Daotang Rivers. The tributaries in the upper parts of the Chaiwan basin now have smaller $\chi$ values than before the capture (Fig. 4a, b) and lower $\chi$ values than the regional trend (Fig. 4d), suggesting that the drainage divide there may eventually migrate to the northwest, further growing the Chaiwen catchment area. In addition, the $\chi$ analysis suggests ongoing drainage divide migration between the upper Daotang River and the Yihe headwaters; lower $\chi$ values in the headwaters of the Daotang River indicate that the Chaiwen catchment continues to grow at the expense of the Yihe River (Fig. 4b, d), which is consistent with our field observations. In this case, ongoing drainage divide migration is accomplished through knickpoint migration (Fig. 2d) that is reversing the flow direction of the upper Yihe River, and adjoining the reversed channel reaches to the Daotang River and the Chaiwen catchment.

## Discussion

We document an extraordinary example of stream capture, which abruptly shifted 25 km² of drainage area from the Yihe to Chaiwen catchments, created the elbow in the Chaiwen River, and incised a ~30 m deep gorge. The capture event likely occurred due to headward extension of the Chaiwen River along the trace of the Shangwujing Fault, and its effects are being felt throughout the Yihe and Chaiwen catchments and neighboring catchments. Following the capture event, both the aggressor and victim drainage basins shifted to a state of topographic disequilibrium, rapid incision, and drainage divide migration. Divide migration and stream piracy is through upstream propagation of a knickpoint, including a trailing 10-m high waterfall, that is actively reversing the water flow direction in a major river. Compared to river basin adjustment by more gradual divide migration alone[6,37,38], river capture has more significant changes for both the aggressor and victim basins, and neighboring catchments, including major stream network reorganization, abrupt drainage area gain and loss, and canyon incision.

## Methods

**Digital Elevation Model (DEM) data**. Figures covering large areas within the study region, such as Figs. 1 and 4 were based on a DEM derived from the ASTER Global DEM Version 2 data[39], which has a resolution of 30 m and a root mean square error (RMSE) of 8.7 m[40].

For the reaches close to the elbow shown in Fig. 2a and all the sub-plots in Supplementary Fig. 1, we used a drone (DJI Phantom 4 Advanced) to take overlapping photographs and then processed them with Agisoft Software to obtain a high-resolution DEM[21] (0.5 m resolution). This DEM was also calibrated using ground check points whose positions were measured using a real-time kinematic Global Navigation Satellite System, with absolute vertical error of 1 m.

**Reconstruction of the topography before the capture event**. To calculate $\chi$ before river capture, it is necessary to reconstruct the planview stream network, the longitudinal elevation profile, and the drainage area of the paleo Chaiwen and Yihe Rivers. For the paleo-Chaiwen River, we used the topography for the present Chaiwen River, but limited the catchment area to be downstream of the elbow. For the paleo Yihe River, we extended the river profile to the headwaters of the Daotang River, following the shortest linear path, and included the Chaiwen River basin upstream of the elbow.

The longitudinal elevation profile along the paleo-Yihe River was reconstructed as follows: firstly, upstream of the present-day Chaiwen knickpoint and along the Yihe River, the longitudinal elevation profiles were assumed to be the same as present; secondly, between the present Chaiwen knickpoint and the wind gap, river-bed elevation was interpolated linearly. The paleo-drainage areas for the paleo-Yihe River were reconstructed as follows: firstly, upstream of the present Chaiwen elbow, the drainage area was assumed to be the same as present; secondly, at the wind gap, the upstream basin area was set as the sum of Daotang River and Chaiwen River upstream of the elbow; and finally, between the elbow and wind gap, drainage area was interpolated linearly.

Although incision likely occurred since the capture in the Yihe River and the Chaiwen River downstream of the elbow, we lack constraints (e.g., terraces) to accurately reconstruct the river elevations in these reaches. The downstream reach of the Chaiwen River lacks an inner gorge, suggesting that incision there may not have been as rapid as upstream of the elbow. Therefore, for the $\chi$ plots we assume that the elevation of the paleo-Chaiwen downstream of the elbow and paleo-Yihe to be the same of the modern rivers, and adjust the drainage areas only. Post-capture incision would shift these river profiles to higher elevations on Fig. 4c.

**The calculation of $\chi$**. Profiles of elevation and $\chi$ can reveal the disequilibrium of river basins resulting from river capture[2,6], in which $\chi$ is defined as

$$\chi = \int_0^x \left(\frac{A_0}{A(x')}\right)^{\frac{m}{n}} dx' \tag{1}$$

where $x$ is the longitudinal distance from the base level point; $A_0$ is a scaling area; $A(x')$ is the upstream basin area at location $x'$; and $m$ and $n$ are exponents applied to the basin area and slope, respectively. We take the values $A_0 = 1\ km^2$ and try different $m/n$ values, and it was found $m/n = 0.45$ corresponded to the best linear fit for the $\chi$-elevation relation of Paleo Yihe and Chaiwen Rivers before capture[36], and is similar to values used elsewhere[13]. The base level for $\chi$ is set to be 299 m, where both the Chaiwen and Yihe Rivers flow from mountain areas to basins, and Chaiwen River also meets the Tongyedian Fault.

**Paleo-flow directions from the dip directions of cobbles**. The dip directions of more than 30 cobbles (64–256 mm in diameter) were measured in each of three sections D1, D2, and D3, with elevation above the present riverbed of 21, 13 and 3 m, respectively (Fig. 2a, b; Supplementary Fig. 2).

**Dating methods and interpretation of the results**. To constrain the age of the river capture event, we used Optically Stimulated Luminescence (OSL) dating measurements, which was carried out by the Beijing OSL Dating Company, China.

We collected two samples, one at D1 containing silt mostly and another at D2 containing sand mostly, and both are just above the imbricated cobbles. The vertical sections were freshly cleaned before sample collections. Steel tubes with 4 cm diameter were wedged horizontally into the layers, and then were wrapped with black plastic bags immediately after they were pulled out.

Samples were prepared under subdued red light (with wave length of 655 ± 30 nm). The two edges of the sample were removed, and ~100 g in the middle portion was processed to extract quartz grains as follows. The materials were treated with 30% $H_2O_2$ for removal of organics. The reminders were treated with 10% HCl for removal of carbonates. The reminders were sieved to get the grains with size fraction of 4–11 and 90–125 μm for samples D1 and D3, respectively. Those grains were etched with 35% $H_2SiF_6$ for about 2 weeks to remove feldspars.

Quartz grains were glued to stainless steel discs with 9.7 mm diameter using silicone oil, and no grains were overlapped. OSL measurements for the quartz were performed with a Risø TL/OSL-DA-20, equipped with an internal $^{90}Sr/^{90}Y$ beta source with blue light ($\lambda = 470 \pm 20$ nm) luminous diode for 60 s stimulation at temperature of 125 °C. The stimulated luminescence signals were detected using a 9235QA photomultiplier tube, fitted with 7.5-mm-thickness Hoya U-340 filters. The concentrations of uranium (U), thorium (Th), and potassium (K) were measured by neutron activation analysis. Dose Rate (DR) can be calculated with concentrations of uranium (U), thorium (Th), and potassium (K) and further calibrated by water content and cosmic radiation[41].

Equal dose (ED) were determined followed Simple Multiple Aliquot-Regenerative Dose method[42]. The age is equal to DR over ED. The ages of the silt in D1 and sand in D3 just above the imbricated cobbles were 82.53 ± 4.29 and 89.48 ± 3.03 ka BP, respectively (see Supplementary Table 1 for the detailed intermediate results). As a result, we consider ~80 ka BP is the maximum age for river capture.

## Data availability

The authors declare that the data supporting the findings of this study are available within the paper and its supplementary files.

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

## Acknowledgements

This study was supported by the National Natural Science Foundation of China (51509172, 51539007, and 41376052), the Open Funding of State Key Laboratory of Loess and Quaternary Geology (No. SKLLQGZR1801) and the personal financial support from Zhen Wu. We thank ASTER GDEM, which is a product of NASA and METI. We thank Roman DiBiase, Huiping Zhang, Rong Yang, Chris Paola, Michael Church, Jeremy G. Venditti, and Tom Dunne for helpful discussions. We are grateful to Tianzhong Li, Zhaopeng Zhang, Ke Zhai, Xiaoyue Fan, Qianbin Sun, Weilai Fan, and Junxiang Kou for their contributions to the field surveys. We thank the reviewers for constructive comments.

## Author contributions

N.F., Z.C., L.J. and M.H. conducted the field surveys. N.F. and M.L. developed the χ analysis. N.F., Z.C., M.H., M.L. and X.L. drafted the paper.

## Additional information

**Competing interests:** The authors declare no competing interests.

