## [Peer Review File · Nature Communications]

Reviewers' comments:

Reviewer #1 (Remarks to the Author):

This paper presents a well-documented case study of a recent river capture event. The argument is based primarily on river channel profiles and map patterns. Map patterns and drainage areas are used to scale channel gradients in order to test how well the capture scenario can explain pre- and post-capture profiles and transient evolution. Additional evidence includes fluvial terrace deposits with paleo-flow indicators demonstrating a reversal of flow direction in precisely the reach as predicted by the model.

The evidence is very convincing. In fact, it is a beautiful example of precisely the set of observations expected, with a rapidly incising channel reach where drainage area has been added, a reversed flow-direction tributary and a drainage-area-starved "victim" river, all of which show the behavior expected for this scenario. The reconstructed drainage pattern and chi plots finish the story with remarkable self-consistency and clarity. The study is clearly presented; the manuscript is well-written and should be published with only a couple issues to be cleared up.

The main problem I see with the analysis is in the interpretation of the knickpoint in the Daotang river. The authors interpret this as a capture related knickpoint and calculate its upstream propagation velocity assuming initiation at the time of capture. However, the knickpoint could not have initiated at the time of capture given that, at this time, there was no Daotang River. At the instant of capture, it still flowed into the Yihe. The reversal of flow direction that resulted in the formation of the Daotang must have developed progressively as the Chaiwen gorge incised at the elbow driving the water divide between the Daotang and the Yihe to the NE. If you allow me to abuse the term "knickpoint", the water divide IS the capture knickpoint. The 10 m high knickpoint identified by the authors must necessarily be a younger feature.

Incidentally, this would be easier to see if the authors would mark the lowest saddle (windgap) between the Daotang and the Yihe on all maps, profiles and chi plots. At the moment, the Daotang profile seems to follow a tributary to a point high above the Yihe channel, which is confusing when trying to visualize where the paleo-Yihe connected across the Daotang.

A second point that could use some clarification if not a recalculation, is how the paleo chi plot is constructed. The methods description is mostly good, but there was no mention of adjusting elevations of the Chaiwen below the capture point at the elbow. There should have been significant incision of the Chaiwen below the capture point, following capture. The authors seem to have corrected the paleo-Yihe for this incision, but the Chaiwen could also be corrected at all points below the "Chaiwen knickpoint". This will require an assumption about how incision rates decrease downstream, but it is constrained by the elevation of the confluence with the Daotang, which must have incised by an amount to bring it up to the paleo-Yihe elevation.

A third point which is a suggestion, not a problem with the current analysis, is with respect to the morphology of the knickpoints and the total incision since capture. It seems like the analysis presented is not taking full advantage of the quality of the data collected and the clarity of the scenario model. With the clear capture sequence, the timing provided by the OSL and the high resolution DEM obtained by the drone, this is a great opportunity to map out and model incision and transient behavior on an 80 ka timescale. There isn't really much presented here; in fact most of this analysis could have been done with a low-res DEM. Some knickpoint celerity rates are given (one of which is probably wrong), but not much more and not much discussion of process or implications for erosion parameters (stream power K at least?). Also the morphology and migration rate of the water

divide, the knickpoints and the channel geometry could be shown and modeled in more detail. The authors could publish the paper with the scope it has, but it feels like a bit of a lost opportunity.

Final comment on style: the last paragraph is rather repetitive and not really needed in such a short paper.

Sean Willett
ETH

Reviewer #3 (Remarks to the Author):

Review: Fan et al., "Abrupt drainage basin reorganization following a Pleistocene river capture in the Yimeng Mountains, China"

Fan et al. document the capture of the Yihe River by the Chaiwen River, which occurred in the Late Pleistocene. Drainage network optimization processes including drainage divide migration and river capture events are in the heart of many research projects. Their potential drivers (tectonic, climate) are discussed in a series of recent high impact publications. However, as mentioned in the manuscript, clear field evidence is rare and therefore extremely valuable to pin down numerical models predicting this process. The applied methods include field mapping, morphometry, and OSL dating. All methods are state of the art and support the story.

Although the study region was under the radar (at least I've never heard of these rivers before), the results and their interpretations are convincing and of broad interest even beyond the geomorphology community. I really enjoyed reading the study. Hence, I suggest to "Invite the authors to revise their manuscript to address specific concerns before a final decision is reached" and justify my suggestion with some minor issues raised by reading the manuscript.

Detailed comments

L 21: "River capture occurs when river network growth outpaces its neighbors, resulting in a victor that steals the neighboring headwaters".

This sentence may be misleading. Somebody may suppose that all catchments (river networks) are growing but some grow faster than others. As the total catchment size remains constant over time, I recommend writing something such as: "River capture may occur during the dynamic reorganization of drainage networks where one catchment grows on the expense of another".

L 33f: ", and active divide migration is propagating the effects of the capture throughout the landscape."

This sentence may be misinterpreted. Of course, the distribution of χ values suggests that the catchment of the Chaiwen River will further grow as consequence of a sudden increase in catchment size and hence run-off. However, the knickpoint, which originated by the capture event, migrated about 1.4 km in upstream direction. This means that the signal of the capture event is still far from the divide. As knickpoints migrate with constant vertical velocity (at least for simple assumptions), it will probably take quite a long time until the signal of the documented capture event will eventually reach the divide and will trigger divide migration (please see [Robl et al., 2017b] and the supplementing videos, 1d-cases).

L 72f: "The longitudinal profiles (Fig. 1c) of the rivers indicate that the divide between the Daotang River and Yihe Rivers is low and flat."

I suppose that the drainage divide is a classical wind gap. However, due to the steep headwater of the Daotang River plotted on Fig. 1c it doesn't look flattish at the first glance. Maybe it would be better to remove the head water section or use a different color code – I don't know.

L 124f: "The relief of the inner gorge decreases upstream from 27 m at the elbow, to the present day knickpoint heights of 2.2 m and 10 m (Extended Data Fig. 1), indicating that the heights of the knickpoints declined as they propagated upstream."

I'm not sure that this statement is fully correct. I'm wondering why the authors assume that the capture event created a huge waterfall (27 m). Maybe I'm wrong but from my understanding, the capture event suddenly increases the discharge of the river downstream the capture event, while the adjustment of the downstream channel towards a new equilibrium (lowering of channel slopes according to the non-linear relationship of channel slope and drainage area described by the stream power law) takes a significant amount of time. However, as a consequence of this downstream channel adjustment, a series of small knickpoints will migrate in upstream direction, which may explain why the gorge is deepest at the capture point (longest time of adjustment). If I'm right than the small waterfalls marking the upstream termination of the gorges represent the "first arrival" of the capture signal (shortest time of adjustment), which started immediately after the capture event (please see [Robl et al., 2017b] and the supplementing videos 2d-cases). In this case, we do not observe a decaying knickpoint but a progressively incising front.

The original "channel profile: Chaiwen - Yihe" doesn't appear well graded. Before the capture event the channel segment of the Chaiwen River upstream the capture point appears "flatter" than the channel segment of the Yihe River. As lithology is uniform and the region was not glaciated I'm just wondering why (spatial variations in uplift rate? active tectonics?). The channel segment downstream the capture point is obviously out of equilibrium, which may easily be explained by the ongoing adjustment of the channel geometry as response to the increased discharge. However, there is a prominent knickpoint at km 11.5. What is the reason for this knickpoint?

L 131: The χ parameter is a metric for weighted drainage area in a river network (Methods) and indicates,

Although working a lot with χ mapping and the χ transform this sentence seems not very explanatory to me.

L 132: "... according to the stream-power model for river incision, disequilibrium in river profiles due to transient adjustment"

In the simplest case this statement is correct. However, in case of spatial variations in uplift rate or erodibility it is not. Please see [Yang et al., 2015; Robl et al., 2017a].

L 154: "suggesting that the drainage divide there is migrating to the northwest, further growing the Chaiwen catchment area."

As the knickpoints are far away from the divides, I would not expect that the divides are already migrating (at least not due to the capture event).

Figures

Fig 1a: Please indicate Yimeng mountains and show the catchments of the Yellow River and the Huaihe River.

The color scales are not really cool. No idea how to convert "yellow" in elevation (1b) or green in χ (4c).

Fig. 4a, c,d: χ should have the units of m or km. In your case I assume km. Don't forget the units of "dx" in equation 1.

I'm looking forward to read this nice study in Nature Communications. I wish you all the best for your revision.

Jörg Robl

Revision notes

We are grateful to your valuable comments. Based on the comments, we have made a revision on the previous version of the manuscript. Our point-by-point responses to the review comments are as the following.

Responses to comments

Reviewer #1 (Sean Willett at ETH):

This paper presents a well-documented case study of a recent river capture event. The argument is based primarily on river channel profiles and map patterns. Map patterns and drainage areas are used to scale channel gradients in order to test how well the capture scenario can explain pre- and post-capture profiles and transient evolution. Additional evidence includes fluvial terrace deposits with paleo-flow indicators demonstrating a reversal of flow direction in precisely the reach as predicted by the model.

The evidence is very convincing. In fact, it is a beautiful example of precisely the set of observations expected, with a rapidly incising channel reach where drainage area has been added, a reversed flow-direction tributary and a drainage-area-starved “victim” river, all of which show the behavior expected for this scenario. The reconstructed drainage pattern and chi plots finish the story with remarkable self-consistency and clarity. The study is clearly presented; the manuscript is well-written and should be published with only a couple issues to be cleared up.

Response: Thanks for the nice comments.

The main problem I see with the analysis is in the interpretation of the knickpoint in the Daotang river. The authors interpret this as a capture related knickpoint and calculate its upstream propagation velocity assuming initiation at the time of capture. However, the knickpoint could not have initiated at the time of capture given that, at this time, there was no Daotang River. At the instant of capture, it still flowed into the Yihe. The reversal of flow direction that resulted in the formation of the Daotang must have developed progressively as the Chaiwen gorge incised at the elbow driving the water divide between the Daotang and the Yihe to the NE. If you allow me to abuse the term “knickpoint”, the water divide IS the capture knickpoint. The 10 m high knickpoint identified by the authors must necessarily be a younger feature.

Response:

We agree with the reviewer and have revised the manuscript. The wind gap is the propagating knickpoint that is reversing the river flow direction. However, the bulk of the incision in Daotang since

capture occurs downstream of a prominent 27 m waterfall that heads an inner gorge and is bounded by terraces. The waterfall likely developed as the river steepened downstream of the wind gap in response to the capture. Because drainage area is negligible at the wind gap itself, the waterfall may reflect a threshold drainage area necessary for significant bedrock incision (DiBiase and Whipple, 2011) or the higher drainage area necessary for waterfall erosion compared to fluvial incision (Scheingross and Lamb, 2017).

Incidentally, this would be easier to see if the authors would mark the lowest saddle (windgap) between the Daotang and the Yihe on all maps, profiles and chi plots. At the moment, the Daotang profile seems to follow a tributary to a point high above the Yihe channel, which is confusing when trying to visualize where the paleo-Yihe connected across the Daotang.

Response: Yes, we have added the wind gap in the figures, i.e., Fig. 1b, Fig. 2a and Fig. 4b.

A second point that could use some clarification if not a recalculation, is how the paleo chi plot is constructed. The methods description is mostly good, but there was no mention of adjusting elevations of the Chaiwen below the capture point at the elbow. There should have been significant incision of the Chaiwen below the capture point, following capture. The authors seem to have corrected the paleo-Yihe for this incision, but the Chaiwen could also be corrected at all points below the “Chaiwen knickpoint”. This will require an assumption about how incision rates decrease downstream, but it is constrained by the elevation of the confluence with the Daotang, which must have incised by an amount to bring it up to the paleo-Yihe elevation.

Response: We agree that there was incision of the Chaiwen below the elbow. Because there are not terraces downstream of the elbow, it is unclear how to reconstruct the paleo-Chaiwen river without making certain assumptions. Importantly there is not an inner gorge downstream of the elbow, unlike upstream of the elbow, suggesting that incision downstream might have been relatively minor (Supplementary Fig. 1). We now make our assumption clear that for the purposes of the χ analysis, we do not attempt to reconstruct the paleo-Chaiwen river elevations (only the drainage area). Any incision that did occur would shift the Paleo-Chaiwen up in elevation on the χ plot (Fig. 4c).

We were unable to reconstruct incision with uncertain erosion rates downstream due to erosion. Accounting for this incision would shift the χ plot for the paleo-chaiwen river up in elevation at Fig 4c, at least at high χ values near the elbow.

A third point which is a suggestion, not a problem with the current analysis, is with respect to the

morphology of the knickpoints and the total incision since capture. It seems like the analysis presented is not taking full advantage of the quality of the data collected and the clarity of the scenario model. With the clear capture sequence, the timing provided by the OSL and the high resolution DEM obtained by the drone, this is a great opportunity to map out and model incision and transient behavior on an 80 ka timescale. There isn't really much presented here; in fact most of this analysis could have been done with a low-res DEM. Some knickpoint celerity rates are given (one of which is probably wrong), but not much more and not much discussion of process or implications for erosion parameters (stream power K at least?). Also the morphology and migration rate of the water divide, the knickpoints and the channel geometry could be shown and modeled in more detail. The authors could publish the paper with the scope it has, but it feels like a bit of a lost opportunity.

Response: Thanks for the suggestion. In the early version of our manuscript, we talked more about channel incision and knickpoint retreat for a simple and condensed paper for Nature Communications. So we focused on the river capture event and the transient adjustment from the χ model. In the new version, we have improved the manuscript for more about the incision and knickpoint retreat.

Final comment on style: the last paragraph is rather repetitive and not really needed in such a short paper.

Response: We have kept the last paragraph and now labeled it as Discussion following the journal conventions.

References

Crosby, B. T. & Whipple, K. X. Knickpoint initiation and distribution within fluvial networks: 236 waterfalls in the Waipaoa River, North Island, New Zealand. *Geomorphology* **82**, 16-38 (2006).

DiBiase, R.A. & Whipple, K.X. The influence of erosion thresholds and runoff variability on the relationships among topography, climate, and erosion rate, *J. Geophys. Res. Earth Surf.* **116**, F04036 (2011).

Scheingross, J. S. & Lamb, M. P. A mechanistic model of waterfall plunge pool erosion into bedrock. *J. Geophys. Res. Earth Surf.* **122**, 2079-2104 (2017).

Reviewer #3 (Jörg Robl):

Fan et al. document the capture of the Yihe River by the Chaiwen River, which occurred in the Late Pleistocene. Drainage network optimization processes including drainage divide migration and river capture events are in the heart of many research projects. Their potential drivers (tectonic, climate) are discussed in a

series of recent high impact publications. However, as mentioned in the manuscript, clear field evidence is rare and therefore extremely valuable to pin down numerical models predicting this process. The applied methods include field mapping, morphometry, and OSL dating. All methods are state of the art and support the story.

Although the study region was under the radar (at least I've never heard of these rivers before), the results and their interpretations are convincing and of broad interest even beyond the geomorphology community. I really enjoyed reading the study. Hence, I suggest to "Invite the authors to revise their manuscript to address specific concerns before a final decision is reached" and justify my suggestion with some minor issues raised by reading the manuscript.

Response: Thanks for the nice comments. Yes, the study region was under the radar. We have added a supplementary kmz file for viewing by Google Earth.

Detailed comments

L 21: "River capture occurs when river network growth outpaces its neighbors, resulting in a victor that steals the neighboring headwaters".

This sentence may be misleading. Somebody may suppose that all catchments (river networks) are growing but some grow faster than others. As the total catchment size remains constant over time, I recommend writing something such as: "River capture may occur during the dynamic reorganization of drainage networks where one catchment grows on the expanse of another".

Response: We agree and have revised the sentence according to the suggestion.

L 33f: ", and active divide migration is propagating the effects of the capture throughout the landscape."

This sentence may be misinterpreted. Of course, the distribution of χ values suggests that the catchment of the Chaiwen River will further grow as consequence of a sudden increase in catchment size and hence runoff. However, the knickpoint, which originated by the capture event, migrated about 1.4 km in upstream direction. This means that the signal of the capture event is still far from the divide. As knickpoints migrate with constant vertical velocity (at least for simple assumptions), it will probably take quite a long time until the signal of the documented capture event will eventually reach the divide and will trigger divide migration (please see [Robl et al., 2017b] and the supplementing videos, 1d-cases).

Response: Thanks for pointing out this inconsistency. As Reviewer 1 Dr Willett noted, the wind gap is the leading knickpoint. In this case, the wind gap knickpoint is actively shifting the divide between the Reversed and Yihe Rivers. The revised version of the manuscript makes this clear.

L 72f: “The longitudinal profiles (Fig. 1c) of the rivers indicate that the divide between the Daotang River and Yihe Rivers is low and flat.”

I suppose that the drainage divide is a classical wind gap. However, due to the steep headwater of the Daotang River plotted on Fig. 1c it doesn't look flattish at the first glance. Maybe it would be better to remove the head water section or use a different color code – I don't know.

Response: Yes, the drainage divide is a classical wind gap. We have removed the headwater section of the Daotang River following the suggestion.

L 124f: “The relief of the inner gorge decreases upstream from 27 m at the elbow, to the present day knickpoint heights of 2.2 m and 10 m (Extended Data Fig. 1), indicating that the heights of the knickpoints declined as they propagated upstream.”

I'm not sure that this statement is fully correct. I'm wondering why the authors assume that the capture event created a huge waterfall (27 m). Maybe I'm wrong but from my understanding, the capture event suddenly increases the discharge of the river downstream the capture event, while the adjustment of the downstream channel towards a new equilibrium (lowering of channel slopes according to the non-linear relationship of channel slope and drainage area described by the stream power law) takes a significant amount of time. However, as a consequence of this downstream channel adjustment, a series of small knickpoints will migrate in upstream direction, which may explain why the gorge is deepest at the capture point (longest time of adjustment). If I'm right than the small waterfalls marking the upstream termination of the gorges represent the “first arrival” of the capture signal (shortest time of adjustment), which started immediately after the capture event (please see [Robl et al., 2017b] and the supplementing videos 2d-cases). In this case, we do not observe a decaying knickpoint but a progressively incising front.

Response: We agree that there should have been incision of the Chaiwen below the elbow, and that the waterfall need not accommodate all of the incision of the gorge. The sentence in question has been removed.

The original “channel profile: Chaiwen - Yihe” doesn't appear well graded. Before the capture event the channel segment of the Chaiwen River upstream the capture point appears “flatter” than the channel segment of the Yihe River. As lithology is uniform and the region was not glaciated I'm just wondering why (spatial variations in uplift rate? active tectonics?). The channel segment downstream the capture point is obviously out of equilibrium, which may easily be explained by the ongoing adjustment of the channel geometry as response

to the increased discharge. However, there is a prominent knickpoint at km 11.5. What is the reason for this knickpoint?

Response: Yes, spatial variations in uplift rate may be the reason of the flatter segment upstream the capture point. We do not have a certain answer for this question, but we do not believe that this point affects our main findings.

L 131: The χ parameter is a metric for weighted drainage area in a river network (Methods) and indicates. Although working a lot with χ mapping and the χ transform this sentence seems not very explanatory to me.

Response: We agree, the more detailed information are now referred to in the “Methods”.

L 132: “... according to the stream-power model for river incision, disequilibrium in river profiles due to transient adjustment”

In the simplest case this statement is correct. However, in case of spatial variations in uplift rate or erodibility it is not. Please see [Yang et al., 2015; Robl et al., 2017a].

Response: We have added “for areas with uniform distributed lift rates and lithology” following “according to the stream-power model for river incision”.

L 154: “suggesting that the drainage divide there is migrating to the northwest, further growing the Chaiwen catchment area.”

As the knickpoints are far away from the divides, I would not expect that the divides are already migrating (at least not due to the capture event).

Response: Yes, we agree, we have rephrased this sentence to indicate that divide migration may eventually occur (at least as indicated by the χ analysis).

Figures

Fig 1a: Please indicate Yimeng mountains and show the catchments of the Yellow River and the Huaihe River.

Response: Yes, revised. We have marked the Yellow River and Huaihe River without showing the catchments for the proper layout of the Fig.1a.

The color scales are not really cool. No idea how to convert “yellow” in elevation (1b) or green in χ

(4c).

Response: We have improved Fig. 1b, Fig. 2a and Fig. 4.

Fig. 4a, c,d: χ should have the units of m or km. In your case I assume km. Don't forget the units of "dx" in equation 1.

Response: Yes, revised.

I'm looking forward to read this nice study in Nature Communications. I wish you all the best for your revision.

Response: Thanks.

References

Mackey, B. H., Scheingross, J. S., Lamb, M. P. & Farley, K. A., Knickpoint formation, rapid propagation, and landscape response following coastal cliff retreat at the last interglacial sea-level highstand: Kaua'i, Hawai'i. *Geol. Soc. Am. Bull.* **126**, 925-942 (2014).

Robl, J., Hergarten, S., & Prasicek, G. The topographic state of fluvially conditioned mountain ranges. *Earth-Sci. Rev.* **168**, 190-217 (2017a).

Robl, J., Heberer, B., Prasicek, G., Neubauer, F., & Hergarten, S. The topography of a continental indenter: The interplay between crustal deformation, erosion, and base level changes in the eastern Southern Alps. *J. Geophys. Res. Earth Surf.* **122**(1), 310-334 (2017b).

Crosby, B. T. & Whipple, K. X. Knickpoint initiation and distribution within fluvial networks: 236 waterfalls in the Waipaoa River, North Island, New Zealand. *Geomorphology* **82**, 16-38 (2006).

Sincerely,

Niannian Fan, Zhongxin Chu, Luguang Jiang, Marwan A. Hassan, Michael P. Lamb and Xingnian Liu

REVIEWERS' COMMENTS:

Reviewer #1 (Remarks to the Author):

The authors have done a good job addressing my original comments and I have no additional comments.

Reviewer #3 (Remarks to the Author):

Review – Revision I: Fan et al., “Abrupt drainage basin reorganization following a Pleistocene river capture in the Yimeng Mountains, China”

I again enjoyed reading this manuscript. The authors have thoroughly implemented the suggestions of my last review. I suggest accepting the manuscript for publication in Nature Communications.

Congratulations

Jörg Robl

I found two minor issues that should be corrected.

L151-152: should be :“ uniformly distributed uplift rates”

L153- : “Before the capture event, the χ value between the headwaters of the paleo-Chaiwen River and the eventual capture point along the paleo Yihe Rivers were similar (Fig. 4a), indicating no significant disequilibrium, and thus χ analysis does not appear to reveal an eventual capture event.”

Looking at Fig. 4a, I do not fully agree. There is a difference in χ across the divides. Fig. 4a does not provide information on the altitude of the two channels but it seems that χ in Fig. 4a predicts the capture event.

Responses to comments

Reviewer #1:

The authors have done a good job addressing my original comments and I have no additional comments.

Response: Thanks for the nice comments.

Reviewer #3 (Jörg Robl):

Review – Revision I: Fan et al., “Abrupt drainage basin reorganization following a Pleistocene river capture in the Yimeng Mountains, China”

I again enjoyed reading this manuscript. The authors have thoroughly implemented the suggestions of my last review. I suggest accepting the manuscript for publication in Nature Communications.

Congratulations

Jörg Robl

Response: Thanks for the nice comments.

I found two minor issues that should be corrected.

L151-152: should be :“ uniformly distributed uplift rates“

Response:

According to the comments, “uniform distributed lift rates” has been changed to be “uniformly distributed uplift rates” in the new manuscript.

L153- : “Before the capture event, the χ value between the headwaters of the paleo-Chaiwen River and the eventual capture point along the paleo Yihe Rivers were similar (Fig. 4a), indicating no significant disequilibrium, and thus χ analysis does not appear to reveal an eventual capture event.”

Looking at Fig. 4a, I do not fully agree. There is a difference in χ across the divides. Fig. 4a does not provide information on the altitude of the two channels but it seems that χ in Fig. 4a predicts the capture event.

The main problem I see with the analysis is in the interpretation of the knickpoint in the Daotang river. The authors interpret this as a capture related knickpoint and calculate its upstream propagation velocity assuming initiation at the time of capture. However, the knickpoint could not have initiated at the time of capture given

that, at this time, there was no Daotang River. At the instant of capture, it still flowed into the Yihe. The reversal of flow direction that resulted in the formation of the Daotang must of developed progressively as the Chaiwen gorge incised at the elbow driving the water divide between the Daotang and the Yihe to the NE. If you allow me to abuse the term “knickpoint”, the water divide IS the capture knickpoint. The 10 m high knickpoint identified by the authors must necessarily be a younger feature.

Response:

In the new manuscript, the two sentences “Before the capture event, the χ value between the headwaters of the paleo-Chaiwen River and the eventual capture point along the paleo Yihe Rivers were similar (Fig. 4a), indicating no significant disequilibrium, and thus χ analysis does not appear to reveal an eventual capture event. Nonetheless, headward extension of the Chaiwen River, following a linear path in χ -elevation space, does intersect the eventual capture point (Fig. 4c), supporting our interpretation.” have been modified to be ‘Before the capture event, the χ -elevation relations of both Paleo Chaiwen and Yihe Rivers generally fitted the regional trend, indicating no significant disequilibrium. Nonetheless, headward extension of the Chaiwen River, following a line ar path in χ -elevation space, does intersect the eventual capture point (Fig. 4c), predicting an eventual capture event.’

Sincerely,

Niannian Fan, Zhongxin Chu, Luguang Jiang, Marwan A. Hassan, Michael P. Lamb and Xingnian Liu